# Reinforcement Learning-Aided Channel Estimator in Time-Varying MIMO Systems

**DOI:** 10.3390/s23125689

**Published:** 2023-06-18

**Authors:** Tae-Kyoung Kim, Moonsik Min

**Affiliations:** 1Department of Electronic Engineering, Gachon University, Seongnam 13120, Republic of Korea; tk415kim@gmail.com; 2School of Electronic and Electrical Engineering, Kyungpook National University, Daegu 41566, Republic of Korea

**Keywords:** data-aided channel estimation, non-iterative approach, first-order Gaussian—Markov channel model, reinforcement learning

## Abstract

This paper proposes a reinforcement learning-aided channel estimator for time-varying multi-input multi-output systems. The basic concept of the proposed channel estimator is the selection of the detected data symbol in the data-aided channel estimation. To achieve the selection successfully, we first formulate an optimization problem to minimize the data-aided channel estimation error. However, in time-varying channels, the optimal solution is difficult to derive because of its computational complexity and the time-varying nature of the channel. To address these difficulties, we consider a sequential selection for the detected symbols and a refinement for the selected symbols. A Markov decision process is formulated for sequential selection, and a reinforcement learning algorithm that efficiently computes the optimal policy is proposed with state element refinement. Simulation results demonstrate that the proposed channel estimator outperforms conventional channel estimators by efficiently capturing the variation of the channels.

## 1. Introduction

The multi-input multi-output (MIMO) system is a key technology in modern communication and can significantly improve channel capacity and communication reliability by using multiple antennas [1,2,3,4,5,6,7]. Spatial multiplexing and diversity gain are representative schemes for this improvement [1,2]. Notably, the channel capacity increases linearly with the number of either transmitter and receiver antennas. However, this increase is based on the unrealistic assumption of perfect channel state information (PCSI) at both the transmitter and receiver.

Many studies have proposed improving the channel estimation accuracy with limited time and frequency resources [8,9,10,11,12,13,14,15]. A representative method is pilot-aided channel estimation, which exploits the information shared between a transmitter and receiver. Linear minimum-mean square-error (LMMSE) channel estimation is a well-known method for pilot-aided channel estimation, owing to its simple structure [8]. However, LMMSE channel estimation exhibits unsatisfactory performance with a limited number of pilots. Thus, many pilots are required to satisfy the performance requirement, which decreases the spectral efficiency.

To overcome this problem, data-aided channel estimation has been investigated in which the detected data symbols are exploited as additional pilot symbols [16,17,18,19,20,21,22,23,24,25,26]. However, the detected data symbols may have errors that degrade the accuracy of channel estimation. The iterative turbo equalizer can overcome this degradation by increasing the maximum-a-posteriori probability (MAP) [16,17,18,19,20,21,22]. However, such an iterative turbo equalizer has considerable complexity and latency at the receivers.

As a non-iterative approach, the reinforcement learning (RL)-aided channel estimator was introduced in [27,28,29,30,31,32,33]. The basic concept of this approach is the sequential selection of detected data symbols to minimize the channel estimation errors. Hence, a Markov decision process (MDP) was defined to solve the sequential selection, and the corresponding optimal policy was derived in a closed-form expression in [31]. In [32], a low-complexity algorithm was investigated by introducing sub-blocks and finite backup samples, and the computational complexity and latency were significantly reduced without performance loss. Recently, a general framework for RL-aided channel estimation was studied in [33] based on Monte Carlo tree search. However, the RL-aided channel estimators in [31,32,33] were originally considered in time-invariant channels; they perform insufficiently in time-varying channels.

In this paper, we propose an RL-aided channel estimator for time-varying MIMO channels. To achieve this, we first introduce an optimization problem for an RL-aided channel estimator in time-varying channels. We then formulate an MDP to solve the optimization problem, and propose an RL algorithm for the MDP that considers the time-varying nature of the channel. The main contributions of this paper are as follows:We propose an RL-aided channel estimator for time-varying channels modeled using a first-order Gaussian—Markov process. First, we define the optimization problem in time-varying channels to select the detected data symbols and minimize the estimation error between the estimated and current channels. This optimization problem is different from those in [31,32,33], where the selection of the detected data symbols is unchanged because the current channel remains unchanged with the time slot index.We propose an RL algorithm for the optimization problem that captures the time-varying nature of a channel. Because the optimization problem minimizes the estimation error between the estimated and current channels, we adjust the weights of the data symbols to improve the channel accuracy of the current channel. Using this adjustment, we derive the optimal policy as a closed-form solution. Note that the proposed optimal policy differs from those in [31,32,33] because the influence of soft-decision symbols in the virtual state for future rewards gradually diminishes as the time slot index increases.We propose a further performance improvement scheme to refine the state elements. This is because the previously selected data symbol degrades the estimation accuracy of the current channel. To improve the estimation accuracy, we refine the previously selected data symbol by reflecting the channel variation. In addition, we remove selected data symbols that are too old by introducing a sliding window, because they have a large noise variance to estimate the current channel. Through simulations, we demonstrated the effectiveness of the proposed channel estimator compared with conventional channel estimators in time-varying channels.

The remainder of this paper is organized as follows. In Section 1, we introduce the system model, optimization problem, and the MDP. The proposed channel estimator, which determines the optimal policy for time-varying channels, is described in Section 2. We propose a further performance improvement scheme in Section 3. In Section 4, we present simulation results to demonstrate the effectiveness of the proposed channel estimator. Finally, we provide our conclusions in Section 5.

## 2. Preliminaries

This section describes the system model of a data-aided channel estimator for time-varying MIMO channels. We present the considered channel estimation and data detection schemes based on the model and introduce an optimization problem for data-aided channel estimation.

### 2.1. System Model

We consider MIMO systems in which a transmitter with a number of transmit antennas $N_{t}$ communicates with a receiver with a number of receive antennas $N_{r}$ (Figure 1). The information is first encoded and mapped to the symbol constellation where X is the symbol constellation set. The transmitted symbol at time *n* denoted by $x\left[n\right]\in X^{N_{t}}$ is then sent over a wireless channel. We model the wireless channel using a first-order Gaussian—Markov process as a time-varying channel model [34,35,36,37,38], where the channel matrix $H\left[n\right]\in C^{N_{t}\times N_{r}}$ has its (t,r)-th component between the *t*-th and *r*-th antennas following a Rayleigh fading CN(0,1) distribution. The temporal correlation of the wireless channel, denoted by ϵ∈[0,1], increases with velocity. Based on this model, the channel matrix H[n] at time slot *n* is given by
(1)$$\begin{array}{cc} H[n] & =\sqrt{1-\epsilon^{2}}H\left[n-1\right]+\epsilon \Delta \left[n\right], \end{array}$$
where Δ[n] follows a CN(0,1) distribution.

When the transmitter sends the symbol x[n] to the receiver over the wireless channel H[n], the received symbol z[n] is given by
(2)$$\begin{array}{cc} z[n] & =H^{H}\left[n\right]x\left[n\right]+n\left[n\right], \end{array}$$
where ${\left(\cdot \right)}^{H}$ denotes the conjugate transpose. n[n] is the additive white Gaussian noise (AWGN) at time slot *n*, with distribution $\mathrm{CN}(0_{N_{r}},\sigma_{n}^{2}I_{N_{r}})$, where $0_{m}$ and $I_{m}$ respectively denote m×m zero and identity matrices.

The frame consists of one pilot and $M_{d}$ data blocks (Figure 1). The pilot block contains $N_{p}$ symbols, whereas each data block contains $N_{d}$ symbols. $M_{p}=\left\{1,\ldots ,N_{p}\right\}$ is defined as the pilot index set and $M_{d}=\left\{\left(d-1\right)N_{d}+1,\ldots ,dN_{d}\right\}$ is defined as the data index set. We consider data-aided channel estimation, where the receiver obtains the initial channel estimates using pilot symbols, and the accuracy of the initial channel estimates is improved by exploiting data symbols.

We adopt the LMMSE method as the basic channel estimation method because it has a simple structure and provides a reasonable performance. Based on the LMMSE method, ${\hat{h}}_{r}$ of the *r*-th row for the initial channel estimate $\hat{H}$ can be obtained as
(3)$$\begin{array}{cc} {\hat{h}}_{r} & ={\left(X^{p}{\left(X^{p}\right)}^{H}+\sigma_{n}^{2}I_{N_{t}}\right)}^{-1}X^{p}{\left(z_{r}^{p}\right)}^{H}, \end{array}$$
where ${\left(\cdot \right)}^{-1}$ is the inverse operation. $X^{p}={\left[x\left[1\right],\ldots ,x\left[N_{p}\right]\right]}^{T}$ and $z_{r}^{p}={\left[z_{r}\left[1\right],\ldots ,z_{r}\left[N_{p}\right]\right]}^{T}$ are the pilot and corresponding received symbols in the pilot block, respectively.

The conventional channel estimator performs data detection at the receiver using the initial channel estimates $\hat{h}$. Because the MAP rule guarantees optimal performance, we adopt it for data detection, which is given by
(4)$$\begin{array}{cc} \hat{x}\left[n\right] & =\underset{x_{k}\in X^{N_{t}}}{\mathrm{argmax}}\theta_{k}\left[n\right]. \end{array}$$
where |·| is the cardinality of a set. $x_{k}\in X^{N_{t}}$ where *k* belongs to the index set of the symbol vector candidate $K=\{1,\ldots ,|X^{N_{t}}|\}$. $\theta_{k}\left[n\right]$ denotes a posteriori probability (APP), which is given by
(5)$$\begin{array}{cc} \theta_{k}\left[n\right] & =\frac{P\left[z\left[n\right]|x\left[n\right]=x_{k}\right]P\left[x\left[n\right]=x_{k}\right]}{\sum_{j\in K}P\left[z\left[n\right]|x\left[n\right]=x_{j}\right]P\left[x\left[n\right]=x_{j}\right]}, \end{array}$$
where the likelihood probability in (5) is calculated by assuming the AWGN channel as
(6)$$\begin{array}{cc} P\left[z\left[n\right]|x\left[n\right]=x_{k}\right] & =\frac{1}{{\left(\pi \sigma_{n}^{2}\right)}^{N_{r}}}e^{-\frac{∥z\left[n\right]-{\hat{h}}^{H}x_{k}{∥}^{2}}{\sigma_{n}^{2}}}. \end{array}$$
where ${∥\cdot ∥}^{2}$ denotes the norm operation and P(·) is the probability of an event. The a priori probability in (5) is also assumed to have an equal probability for possible candidate transmitted symbol $P\left[x\left[n\right]=x_{k}\right]=\frac{1}{{\left|X\right|}^{N_{t}}}$.

### 2.2. Problem

In a time-varying channel, the estimation accuracy of $\hat{h}$ decreases gradually as time slot index *n* increases. This degradation results in poor detection performance at the receiver. Because the detected data symbol may have an error owing to the channel, an incorrect use of the detected data symbol severely degrades performance. To overcome this degradation, we consider a data-aided channel estimator that selects the detected data symbols for data-aided channel estimation.

For the selection, we define action a∈A={0,1} where the detected data symbol is used in channel estimation when a=1; otherwise, the detected data symbol is not used. When we define $a\in {\left\{0,1\right\}}^{N_{d}}$ as a set of actions, the considered data-aided channel estimation can be obtained using this set as
(7)$$\begin{array}{cc} {\hat{h}}_{r}\left(a\right) & ={\left(\hat{X}\left(a\right){\hat{X}}^{H}\left(a\right)+\sigma_{n}^{2}I_{N_{t}}\right)}^{-1}\hat{X}\left(a\right)z_{r}^{H}\left(a\right) \end{array}$$
where $z_{r}\left(a\right)={\left[z_{r}^{p},z_{r}\left[e_{1}\left(a\right)\right],\ldots ,z_{r}\left[e_{{∥a∥}_{0}}\left(a\right)\right]\right]}^{T}$ and $\hat{X}\left(a\right)={\left[X^{p},\hat{x}\left[e_{1}\left(a\right)\right],\ldots ,\hat{x}\left[e_{{∥a∥}_{0}}\left(a\right)\right]\right]}^{T}$. The time slot index of the *i*-th nonzero element is denoted as $e_{i}\left(\cdot \right)$. We then define the optimization problem as
(8)$$\begin{array}{cc} a^{★} & =\underset{a}{\mathrm{argmin}}E\{∥\hat{h}\left(a\right)-H\left[n\right]{∥}^{2}\}, \end{array}$$
where E(·) is the expectation of a random variable.

Compared with previous studies [31,32,33], the optimization problem in (8) considers the selection to minimize the MSE between the estimated channel and H[n]. Because the channel is variant with time slot index *n*, the best action $a^{★}$ may be different with time slot index *n*. That is, the best action in the previous time slot index may be invalid in the next time slot index. In addition, the optimization problem is difficult to solve because the number of candidate actions increases exponentially with the data symbol length. An exhaustive search for action candidates is not feasible in practical applications. To resolve these difficulties, we introduce a sequential selection of the detected data symbols and a refinement of the selected data.

### 2.3. Markov Decision Process

We formulate an MDP that solves the optimization problem in (8). To achieve this, we define state $S_{n}$, transition function $T_{n+1}^{\left(a,j\right)}\left(S_{n}\right)$, action A, and reward $R(S_{n},S_{n+1})$ [39]. Subsequently, the Q-value function $Q\left(S_{n},a\right)$ and the optimal policy $\pi^{★}\left(S_{n}\right)$ will be presented. The basic definitions for the MDP are adopted from those in [31,32,33]; however, the RL solution for the MDP is different from those in previous studies, which will be explained in the next section.

The state set $S_{n}$ is defined as
(9)$$\begin{array}{cc} S_{n}=\{\left(X_{n},{\hat{X}}_{n},C\right)| & X_{n}=\left[x\left[1\right]\cdots x\left[N_{p}\right],x_{k_{C\left(1\right)}},\cdots ,x_{k_{C\left(|C|\right)}}\right],\\ & {\hat{X}}_{n}=\left[x\left[1\right]\cdots x\left[N_{p}\right],\hat{x}\left[C\left(1\right)\right],\cdots ,\hat{x}\left[C\left(|C|\right)\right]\right],\\ & C\subset \left\{1,\cdots ,n-1\right\}\}, \end{array}$$
where C is the set of time slot indices where the symbol is used in channel estimation, and C(i) is the *i*-th smallest element. $k_{n}\in K=\left\{1,\ldots ,|X|\right\}$ is the transmitted symbol index at time slot *n*. Based on the expression, we can obtain the proposed channel estimate using the state $S_{n}\in S_{n}$ as
(10)$$\begin{array}{cc} {\hat{h}}_{r}\left(S_{n}\right) & ={\left({\hat{X}}_{n}{\hat{X}}_{n}^{H}+\sigma_{n}^{2}I_{N_{t}}\right)}^{-1}{\hat{X}}_{n}z_{r}^{H}\left(S_{n}\right) \end{array}$$
where $z_{r}\left(S_{n}\right)={\left[z_{r}^{p},z_{r}\left[C\left(1\right)\right],\ldots ,z_{r}\left[C\left(|C|\right)\right]\right]}^{T}$. Note that $S_{n}$ is the set of all states and $S_{n}$ is the state.

The action set is defined as A={0,1}. As explained in the previous subsection, the detected data symbol is used in the proposed channel estimation when a=1; otherwise, the detected data symbol is not used. The transition function $T^{\left(a,j\right)}\left(S_{n}\right)$ from state $S_{n}\in S_{n}$ is defined as
(11)$$\begin{array}{cc} T^{\left(a,j\right)}\left(S_{n}\right) & =P\left[U_{n+1}^{\left(a,j\right)}\left(S_{n}\right)|S_{n},a\right]=\left\{\begin{array}{cc} I\left[x\left[n\right]=x_{j}\right], & j\in J_{a},a=1,\\ 1, & j\in J_{a},a=0. \end{array}\right. \end{array}$$
where I(·) equals one when the event is true and zero otherwise. $J_{0}\in =\left\{0\right\}$ and $J_{1}\in \left\{1,\ldots ,K\right\}$. $U_{n+1}^{\left(a,j\right)}\left(S_{n}\right)\in U_{n+1}^{\left(a,j\right)}\left(S_{n}\right)$ is a possible candidate for the next state from state $S_{n}$ and is defined as
(12)$$\begin{array}{cc} U_{n+1}^{\left(a,j\right)}\left(S_{n}\right) & =\left\{\begin{array}{cc} \left(\left[X_{n},x_{j}\right],\left[{\hat{X}}_{n},\hat{x}\left[n\right]\right],\left[C\cup n\right]\right), & j\in J_{a},a=1,\\ \left(X_{n},{\hat{X}}_{n},C\right), & j\in J_{a},a=0. \end{array}\right. \end{array}$$

The reward $R\left(S_{n},S_{n+1}\right)$ is defined as the difference between the MSEs at the current state $S_{n}\in S_{n}$ and the next state $S_{n+1}\in S_{n+1}$, which is given by
(13)$$\begin{array}{cc} R\left(S_{n},S_{n+1}\right) & =E\left[∥{\hat{h}}_{r}\left(S_{n}\right)-h_{r}{\left[n\right]∥}^{2}\right]-E\left[∥{\hat{h}}_{r}\left(S_{n+1}\right)-h_{r}{\left[n+1\right]∥}^{2}\right]\\ & =\mathrm{Tr}\left[B\left(S_{n}\right)\right]-\mathrm{Tr}\left[B\left(S_{n+1}\right)\right]=\mathrm{Tr}\left[B\left(S_{n}\right)-B\left(S_{n+1}\right)\right], \end{array}$$
where $B\left(S_{n}\right)=E\left[\left({\hat{h}}_{r}\left(S_{n}\right)-h_{r}\left[n\right]\right){\left({\hat{h}}_{r}\left(S_{n}\right)-h_{r}\left[n\right]\right)}^{H}\right]$ is error covariance. Unlike in [31,32,33], the error covariance is defined between the estimated channel ${\hat{h}}_{r}\left(S_{n}\right)$ and $h_{r}\left[n\right]$ at time slot index *n*.

The Q-value function $Q\left(S_{n},a\right)$ is the sum of the rewards, which is given by
(14)$$\begin{array}{cc} Q\left(S_{n},a\right) & =\sum_{j\in J_{a}}T^{\left(a,j\right)}\left(S_{n}\right)\left[R\left(S_{n},U_{n+1}^{\left(a,j\right)}\left(S_{n}\right)\right)+\gamma V^{★}\left(U_{n+1}^{\left(a,j\right)}\left(S_{n}\right)\right)\right], \end{array}$$
where Tr(·) is a trace operation. $V^{★}\left(U_{n+1}^{\left(a,j\right)}\left(S_{n}\right)\right)$ is the optimal sum of future reward after $U_{n+1}^{\left(a,j\right)}\left(S_{n}\right)$. γ is a discounting factor whose value is assumed as one because the proposed channel estimator also considers the effect of future rewards at the ending state [31].

The optimal policy maximizes the Q-value function, which is expressed as
(15)$$\begin{array}{cc} \pi^{★}\left(S_{n}\right) & =\underset{a\in A}{\mathrm{argmax}}Q\left(S_{n},a\right). \end{array}$$

Solving the optimization problem in (15) is highly difficult because the transition probability $T^{\left(a,j\right)}\left(S_{n}\right)$ is unknown, and the number of candidate states exponentially increases with the data length. An effective method to solve this problem is to use a reinforcement learning algorithm. Therefore, the proposed channel estimator also adopts a reinforcement learning algorithm, but the effect of the time-varying channel is also considered in comparison with [31,32,33].

A deep reinforcement learning (DRL) approach is a promising solution for dealing with the dimension explosion of the states by leveraging deep neural networks. To apply the DRL approach to our MDP, an agent needs to interact with an environment to obtain an action-value function for a given action and state. However, both the states and rewards of our MDP are not observable at the receiver. This means that the agent cannot acquire training samples, each of which consists of the state (or the state transition) and the corresponding reward. Consequently, the DRL approach and other data-driven approaches are not directly applicable to solving our MDP.

## 3. Proposed Optimal Policy

This section describes the proposed optimal policy. The basic concept of the derivation is similar to that in [31,32,33]. However, its direct extension is difficult for time-varying channels. This is because capturing time-variant channels using previously selected data symbols is difficult. To address this, we approximate the first-order Gaussian—Markov process and propose a computationally efficient algorithm.

We employ the approximation in [31,32,33] for the transition function, which is given by
(16)$$\begin{array}{cc} {\hat{T}}^{\left(a,j\right)}\left(S_{n}\right) & =\left\{\begin{array}{cc} \theta_{j}\left[n\right], & j\in J_{a},a=1,\\ 1, & j\in J_{a},a=0, \end{array}\right. \end{array}$$
where ${\hat{T}}^{\left(a,j\right)}\left(S_{n}\right)\to T^{\left(a,j\right)}\left(S_{n}\right)$ as $\theta_{j}\left[n\right]\to 1$.

The main difficulty in analyzing the time-varying channel model is solving element Δ[n]. To resolve this difficulty, we approximate the first-order Gaussian—Markov process in (1) as follows:(17)$$\begin{array}{cc} H[n] & \approx \sqrt{1-\epsilon^{2}}H\left[n-1\right], \end{array}$$
where H[n−1]≫Δ[n]. This approximation is often adopted in studies because it provides analytical tractableness [36,37,38]. Using this approximation, the received symbol z[n+m] for 1≤m can be expressed in terms of H[n] as follows:(18)$$\begin{array}{cc} z[n+m] & =H^{H}\left[n+m\right]x\left[n+m\right]+n\left[n+m\right]\\ & \approx H^{H}\left[n\right]\left({\sqrt{1-\epsilon^{2}}}^{m}x\left[n+m\right]\right)+n\left[n+m\right], \end{array}$$

From approximation (18), the virtual state in [31] that mimics the optimal behavior from state $U_{n+1}^{\left(a,j\right)}\left(S_{n}\right)$ can be obtained as follows:(19)$$\begin{array}{cc} {\tilde{U}}_{m}^{\left(a,j\right)}\left(S_{n}\right) & =\left(X_{m}^{\left(a,j\right)},{\hat{X}}_{m}^{\left(a\right)},C_{m}^{\left(a\right)}\right), \end{array}$$
where
$$\begin{array}{cc} X_{m}^{\left(a,j\right)} & =\left\{\begin{array}{cc} \left[X_{n},x_{j},\left(\sqrt{1-\epsilon^{2}}\tilde{x}\left[n+1\right]\right),\cdots ,\left({\sqrt{1-\epsilon^{2}}}^{m-n-1}\tilde{x}\left[m-1\right]\right)\right], & a=1,\\ \left[X_{n},\left(\sqrt{1-\epsilon^{2}}\tilde{x}\left[n+1\right]\right),\cdots ,\left({\sqrt{1-\epsilon^{2}}}^{m-n-1}\tilde{x}\left[m-1\right]\right)\right], & a=0. \end{array}\right.\\ {\hat{X}}_{m}^{\left(a\right)} & =\left\{\begin{array}{cc} \left[{\hat{X}}_{n},\hat{x}\left[n\right],\left(\sqrt{1-\epsilon^{2}}\tilde{x}\left[n+1\right]\right),\cdots ,\left({\sqrt{1-\epsilon^{2}}}^{m-n-1}\tilde{x}\left[m-1\right]\right)\right], & a=1,\\ \left[{\hat{X}}_{n},\left(\sqrt{1-\epsilon^{2}}\tilde{x}\left[n+1\right]\right),\cdots ,\left({\sqrt{1-\epsilon^{2}}}^{m-n-1}\tilde{x}\left[m-1\right]\right)\right], & a=0. \end{array}\right.\\ C_{m}^{\left(a\right)} & =\left\{\begin{array}{cc} \left[C\cup \{n+1,\ldots ,m\}\right], & a=1,\\ \left[C\cup \{n+2,\ldots ,m\}\right], & a=0. \end{array}\right. \end{array}$$

The soft-decision symbol $\tilde{x}\left[m\right]$ for m≥n+1 is define as
(20)$$\begin{array}{cc} \tilde{x}\left[m\right] & =\sum_{k=1}^{K}\theta_{k}\left[m\right]x_{j}. \end{array}$$

In (19), because 0≤ϵ≤1, the effect of soft decision symbol $\tilde{x}\left[n+m\right]$ for estimating H[n] is diminished as *m* increases. Based on the virtual state, the state-action diagram for the proposed channel estimator is shown in Figure 2. In this figure, the number of state transitions at state $S_{n}$ are one and *K* for a=0 and a=1, respectively. However, after n+2, the state transition is simplified to one because the virtual state mimics the behavior of state $U_{n+1}^{\left(a,j\right)}\left(S_{n}\right)$.

Using the definition of virtual state (19), we can compute the future reward $V^{★}\left(U_{n+1}^{\left(a,j\right)}\left(S_{n}\right)\right)$ as
(21)$$\begin{array}{cc} V^{★}\left(U_{n+1}^{\left(a,j\right)}\left(S_{n}\right)\right) & \approx R\left(U_{n+1}^{\left(a,j\right)}\left(S_{n}\right),{\tilde{U}}_{n+2}^{\left(a,j\right)}\left(S_{n}\right)\right)+\sum_{m=n+2}^{M_{d}\left(N_{d}\right)}R\left({\tilde{U}}_{m}^{\left(a,j\right)}\left(S_{n}\right),{\tilde{U}}_{m+1}^{\left(a,j\right)}\left(S_{n}\right)\right). \end{array}$$

By applying (13) to the future reward, the future reward is simplified as
(22)$$\begin{array}{cc} V^{★}\left(U_{n+1}^{\left(a,j\right)}\left(S_{n}\right)\right) & =\mathrm{Tr}\left[B\left(U_{n+1}^{\left(a,j\right)}\left(S_{n}\right)\right)-B\left({\tilde{U}}_{n+2}^{\left(a,j\right)}\left(S_{n}\right)\right)+\sum_{m=n+2}^{M_{d}\left(N_{d}\right)}B\left({\tilde{U}}_{m}^{\left(a,j\right)}\left(S_{n}\right)\right)-B\left({\tilde{U}}_{m+1}^{\left(a,j\right)}\left(S_{n}\right)\right)\right]\\ & =\mathrm{Tr}\left[B\left(U_{n+1}^{\left(a,j\right)}\left(S_{n}\right)\right)-B\left({\tilde{U}}_{M_{d}\left(N_{d}\right)+1}^{\left(a,j\right)}\left(S_{n}\right)\right)\right]. \end{array}$$

Using the approximations (16) and (22), the Q-value function in (14) is obtained as follows:(23)$$\begin{array}{cc} Q\left(S_{n},a\right) & =\sum_{j\in J_{a}}{\hat{T}}^{\left(a,j\right)}\left(S_{n}\right)\mathrm{Tr}\left[B\left(S_{n}\right)-B\left({\tilde{U}}_{M_{d}\left(N_{d}\right)+1}^{\left(a,j\right)}\left(S_{n}\right)\right)\right]. \end{array}$$

The error covariance matrix $B\left({\tilde{U}}_{m}^{\left(a,j\right)}\right)$ can be computed as
(24)$$\begin{array}{cc} B\left({\tilde{U}}_{m}^{\left(a,j\right)}\right) & =E\{∥{\hat{h}}_{r}\left({\tilde{U}}_{m}^{\left(a,j\right)}\right)-h_{r}\left[n\right]{∥}^{2}\}\\ & =E\{\left({\hat{h}}_{r}\left({\tilde{U}}_{m}^{\left(a,j\right)}\right)-h_{r}\left[n\right]\right){\left({\hat{h}}_{r}\left({\tilde{U}}_{m}^{\left(a,j\right)}\right)-h_{r}\left[n\right]\right)}^{H}\}\\ & =E\left\{{\hat{h}}_{r}\left({\tilde{U}}_{m}^{\left(a,j\right)}\right){\hat{h}}_{r}^{H}\left({\tilde{U}}_{m}^{\left(a,j\right)}\right)-h_{r}\left[n\right]{\hat{h}}_{r}^{H}\left({\tilde{U}}_{m}^{\left(a,j\right)}\right)-{\hat{h}}_{r}\left({\tilde{U}}_{m}^{\left(a,j\right)}\right)h_{r}^{H}\left[n\right]+h_{r}\left[n\right]h_{r}^{H}\left[n\right]\right\}\\ & \overset{\left(a\right)}{=}Q_{m}^{\left(a\right)}{\hat{X}}_{m}^{\left(a,j\right)}\left({\left(X_{m}^{\left(a,j\right)}\right)}^{H}X_{m}^{\left(a,j\right)}+\sigma_{n}^{2}I_{|C_{m}^{\left(a\right)}|}\right){\left({\hat{X}}_{m}^{\left(a,j\right)}\right)}^{H}Q_{m}^{\left(a\right)}\\ & -X_{m}^{\left(a,j\right)}{\left({\hat{X}}_{m}^{\left(a,j\right)}\right)}^{H}Q_{m}^{\left(a\right)}-Q_{m}^{\left(a\right)}{\hat{X}}_{m}^{\left(a,j\right)}{\left(X_{m}^{\left(a,j\right)}\right)}^{H}+I_{N_{t}}\\ & =\left(I_{N_{t}}-Q_{m}^{\left(a\right)}{\hat{X}}_{m}^{\left(a,j\right)}{\left(X_{m}^{\left(a,j\right)}\right)}^{H}\right){\left(I_{N_{t}}-Q_{m}^{\left(a\right)}{\hat{X}}_{m}^{\left(a,j\right)}{\left(X_{m}^{\left(a,j\right)}\right)}^{H}\right)}^{H}+\sigma_{n}^{2}Q_{m}^{\left(a\right)}-\sigma_{n}^{4}{\left(Q_{m}^{\left(a\right)}\right)}^{2}\\ & \overset{\left(b\right)}{=}Q_{m}^{\left(a\right)}D_{m}^{\left(a,j\right)}{\left(D_{m}^{\left(a,j\right)}\right)}^{H}Q_{m}^{\left(a\right)}+\sigma_{n}^{2}Q_{m}^{\left(a\right)}-\sigma_{n}^{4}{\left(Q_{m}^{\left(a\right)}\right)}^{2} \end{array}$$
where the distribution of $z_{r}^{H}\left({\tilde{U}}_{m}^{\left(a,j\right)}\left(S_{n}\right)\right)$ is given by $\mathrm{CN}\left(0_{|C_{m}^{\left(a\right)}|},{\left(X_{m}^{\left(a,j\right)}\right)}^{H}X_{m}^{\left(a,j\right)}+\sigma_{n}^{2}I_{|C_{m}^{\left(a\right)}|}\right)$ and $Q_{m}^{\left(a\right)}={\left({\hat{X}}_{n}{\hat{X}}_{n}^{H}+\sum_{k=n+1}^{m-1}{\left(1-\epsilon^{2}\right)}^{m-n}\tilde{x}\left[k\right]{\tilde{x}}^{H}\left[k\right]+\sigma_{n}^{2}I_{N_{t}}\right)}^{-1}$ is applied in (a). $D_{m}^{\left(a,j\right)}={\left(Q_{m}^{\left(a\right)}\right)}^{-1}-{\hat{X}}_{m}X_{m}^{H}={\hat{X}}_{m}{\left({\hat{X}}_{m}-X_{m}\right)}^{H}+\sigma_{n}^{2}I_{N_{t}}$ is used in (b).

By applying (24) to the Q-value function, the optimal policy at $S_{n}$ is computed as
(25)$$\begin{array}{cc} \pi^{★}\left(S_{n}\right) & =\underset{a\in \{0,1\}}{\mathrm{argmax}}Q\left(S_{n},a\right)=I\left[\left(Q\left(S_{n},1\right)-Q\left(S_{n},0\right)\right)\geq 0\right]\\ & =I\left[\mathrm{Tr}\left[\left(\sum_{j=1}^{K}B\left({\tilde{U}}_{M_{d}\left(N_{d}\right)+1}^{\left(0,0\right)}\left(S_{n}\right)\right)-\theta_{j}\left[n\right]B\left({\tilde{U}}_{M_{d}\left(N_{d}\right)+1}^{\left(1,j\right)}\left(S_{n}\right)\right)\right)\right]\geq 0\right]. \end{array}$$
where $B\left({\tilde{U}}_{M_{d}\left(N_{d}\right)+1}^{\left(a,j\right)}\left(S_{n}\right)\right)=\sigma_{n}^{2}Q^{\left(a\right)}-\sigma_{n}^{4}{\left(Q^{\left(a\right)}\right)}^{2}+Q^{\left(a\right)}D^{\left(a,j\right)}{\left(D^{\left(a,j\right)}\right)}^{H}Q^{\left(a\right)}$. $Q^{\left(a\right)}=Q_{M_{d}\left(N_{d}\right)+1}^{\left(a\right)}$ and $D^{\left(a,j\right)}=D_{M_{d}\left(N_{d}\right)+1}^{\left(a,j\right)}$ are defined as
$$\begin{array}{cc} Q^{\left(a\right)} & ={\left({\hat{X}}_{M_{d}\left(N_{d}\right)+1}^{\left(a\right)}{\left({\hat{X}}_{M_{d}\left(N_{d}\right)+1}^{\left(a\right)}\right)}^{H}+\sigma_{n}^{2}I_{N_{\mathrm{tx}}}\right)}^{-1}\\ & \overset{\left(a\right)}{=}\left\{\begin{array}{cc} {\left({\hat{X}}_{n}{\hat{X}}_{n}^{H}+\sum_{m=n+1}^{M_{d}\left(N_{d}\right)}{\left(1-\epsilon^{2}\right)}^{m-n}\tilde{x}\left[m\right]{\tilde{x}}^{H}\left[m\right]+\sigma_{n}^{2}I_{N_{t}}\right)}^{-1}, & a=0,\\ {\left({\left(Q^{\left(0\right)}\right)}^{-1}+\tilde{x}\left[n\right]{\tilde{x}}^{H}\left[n\right]\right)}^{-1}, & a=1. \end{array}\right.\\ D^{\left(a,j\right)} & ={\hat{X}}_{M_{d}\left(N_{d}\right)+1}^{\left(a\right)}{\left({\hat{X}}_{M_{d}\left(N_{d}\right)+1}^{\left(a\right)}-X_{M_{d}\left(N_{d}\right)+1}^{\left(a,j\right)}\right)}^{H}+\sigma_{n}^{2}I_{N_{t}}\\ & \overset{\left(b\right)}{=}\left\{\begin{array}{cc} {\hat{X}}_{n}{\left({\hat{X}}_{n}-X_{n}\right)}^{H}+\sigma_{n}^{2}I_{N_{t}}, & j\in J_{a},a=0,\\ D^{\left(0,0\right)}+\hat{x}\left[n\right]{\left(\hat{x}\left[n\right]-x_{j}\right)}^{H}, & j\in J_{a},a=1. \end{array}\right. \end{array}$$

Similar to [31], $Q^{\left(1\right)}$ and $Q^{\left(0\right)}$ satisfy $Q^{\left(1\right)}=Q^{\left(0\right)}-\frac{Q^{\left(0\right)}\hat{x}\left[n\right]{\hat{x}}^{H}\left[n\right]Q^{\left(0\right)}}{1+{\hat{x}}^{H}\left[n\right]Q^{\left(0\right)}\hat{x}\left[n\right]}$. In addition, $D^{\left(1,j\right)}$ and $D^{\left(0,0\right)}$ satisfy $\sum_{j=1}^{K}\theta_{j}\left[n\right]D^{\left(1,j\right)}{\left(D^{\left(1,j\right)}\right)}^{H}=\left(D^{\left(0,0\right)}+{\hat{d}}_{n}\right){\left(D^{\left(0,0\right)}+{\hat{d}}_{n}\right)}^{H}+\delta_{n}\hat{x}\left[n\right]{\hat{x}}^{H}\left[n\right]$ where ${\hat{d}}_{n}=\hat{x}\left[n\right]{\left(\hat{x}\left[n\right]-\tilde{x}\left[n\right]\right)}^{H}$, and $\delta_{n}=\sum_{j=1}^{K}\theta_{j}\left[n\right]∥\hat{x}\left[n\right]-x_{j}{∥}^{2}-{∥\hat{x}\left[n\right]-\tilde{x}\left[n\right]∥}^{2}$.

Finally, similar to [32], by applying the results in (23) and (24) to (25), we obtain the proposed optimal policy in closed-form as
(26)$$\begin{array}{cc} \pi^{★}\left(S_{n}\right) & =I\left[\frac{\sigma_{n}^{2}\left(1+\alpha_{n}\right)+\sigma_{n}^{4}∥a_{n}{∥}^{2}+{∥d_{n}∥}^{2}}{2\sigma_{n}^{4}\beta_{n}+\gamma_{n}+{∥c_{n}-b_{n}+d_{n}∥}^{2}}\geq 1\right]. \end{array}$$

When we define $Q=Q^{\left(0\right)}$ and $D=D^{\left(0,0\right)}$, vectors are computed as $a_{n}=\frac{Q\hat{x}\left[n\right]}{\sqrt{1+\alpha_{n}}}$, $b_{n}=D^{H}b_{n}$, $c_{n}=\frac{\hat{x}\left[n\right]-\tilde{x}\left[n\right]}{\sqrt{1+\alpha_{n}}}$, and $d_{n}=\frac{D^{H}Qa_{n}}{∥a_{n}{∥}^{2}}$. In addition, the constants are computed as $\alpha_{n}={\hat{x}}^{H}\left[n\right]Q\hat{x}\left[n\right]$, $\beta_{n}=\frac{a_{n}^{H}Qa_{n}}{∥a_{n}{∥}^{2}}$, and $\gamma_{n}=\frac{\delta_{n}}{1+\alpha_{n}}$. Note that the expression of the optimal policy in (26) is similar to that in [32]. However, the vectors and constants in the optimal policy is different from those in [32] because the temporal correlation ϵ is considered in Q and D. When ϵ=0, the optimal policy in (26) is equivalent to that in [32].

## 4. Further Performance Improvement

In this section, we propose a practical method to improve the estimation accuracy of the proposed channel estimator. The proposed method refines state elements to capture the time-varying nature of the channel.

### 4.1. State Element Refinement

Elements $X_{n}$ and ${\hat{X}}_{n}$ in state $S_{n}$ are updated when the detected data symbol is selected based on the optimal policy. However, the elements gradually lose their effectiveness in estimating H[n] as time slot index *n* increases. To address this, we first represent the received symbol for 1≤m in terms of H[n] as
(27)$$\begin{array}{cc} z[n-m] & =H^{H}\left[n-m\right]x\left[n-m\right]+n\left[n-m\right]\\ & \approx H^{H}\left[n\right]\left({\sqrt{1-\epsilon^{2}}}^{-m}x\left[n-m\right]\right)+n\left[n-m\right]. \end{array}$$

Using (27), we refine the elements $X_{n}$ and ${\hat{X}}_{n}$ in state as the time slot index increases, which is given by
(28)$$\begin{array}{cc} X_{n} & \leftarrow {\sqrt{1-\epsilon^{2}}}^{-1}X_{n}\\ {\hat{X}}_{n} & \leftarrow {\sqrt{1-\epsilon^{2}}}^{-1}{\hat{X}}_{n}. \end{array}$$

Regardless of the above refinement, the previously selected data symbols lose their effectiveness as the time slot index increases, particularly for large data lengths. This is because the term Δ[n] in (1) becomes dominant, increasing the uncertainty in estimating the channel. To overcome this, we remove too-old selected data symbols in state by introducing a window size $N_{w}$. In other words, we maintain the size of the set of time slot indices as $\left|C\right|=N_{w}$. Thus, when the optimal action is one at time slot index *n*, *n* is included, whereas the first index C(1) is removed from set C, which can be expressed as
(29)$$\begin{array}{cc} X_{n} & \leftarrow X_{n}\setminus X_{n}\left[C\left(1\right)\right],\\ {\hat{X}}_{n} & \leftarrow {\hat{X}}_{n}\setminus {\hat{X}}_{n}\left[C\left(1\right)\right],\\ C & \leftarrow C\setminus C(1). \end{array}$$

### 4.2. Algorithm

Using the proposed optimal policy and performance improvement strategy, the proposed channel estimator is summarized in Algorithm 1. The receiver obtains the initial channel estimation during pilot transmission. Subsequently, during the data transmission, the receiver sequentially selects a data symbol based on the optimal policy. When the optimal action $a^{★}=1$, the state $S_{n}$ is updated using the most-probable state transition [31]. In addition, the state element refinement is performed based on this condition. After each data block ends, the channel estimate is updated using the state $S_{n}$.
**Algorithm 1:** Proposed channel estimator**1** Obtain the initial channel estimate $H\leftarrow \hat{h}=\left[{\hat{h}}_{1},\cdots ,{\hat{h}}_{N_{r}}\right]$ from (3)**2** Initialize the state $S_{1}=\left(X^{p},X^{p},\phi \right)$.**3 for*** d=1 to $M_{d}$ ***do**(**4**   **for**$\;n\in M_{d}$**do**(**5**    Compute the optimal policy $a^{★}=\pi^{★}\left(S_{n}\right)$ from (26).**6**    Set the optimal values $j^{★}=0$ for $a^{★}=0$ and $x_{j^{★}}=\hat{x}\left[n\right]$ for $a^{★}=1$.**7**    Update $S_{n+1}\leftarrow U_{n+1}^{\left(a^{★},j^{★}\right)}\left(S_{n}\right)$ from (12).**8**    **if***$\;a^{★}==1$ and $N_{w}<\left|C\right|$***then**(**9**      Remove the state elements in $S_{n+1}$.**10**      $X_{n+1}\leftarrow X_{n+1}\setminus X_{n+1}\left[C\left(1\right)\right]$,**11**      ${\hat{X}}_{n+1}\leftarrow {\hat{X}}_{n+1}\setminus {\hat{X}}_{n}\left[C\left(1\right)\right]$,**12**      C←C∖C(1).**13**    **end**(**14**    Refine the state elements in $X_{n+1}\leftarrow {\sqrt{1-\epsilon^{2}}}^{-1}X_{n+1}$ and      ${\hat{X}}_{n+1}\leftarrow {\sqrt{1-\epsilon^{2}}}^{-1}{\hat{X}}_{n+1}$.**15**  **end**(**16**  Update the channel estimate $H\leftarrow \hat{h}=\left[{\hat{h}}_{1}\left(S_{n}\right),\cdots ,{\hat{h}}_{N_{r}}\left(S_{n}\right)\right]$ from (10).**17 end**(

In Figure 3, we show a block diagram of the proposed channel estimator, which consists of the LMMSE channel estimator, optimal policy calculator, and state element refinement. The LMMSE channel estimator obtains the initial estimate at pilot transmission and updates the estimate at data transmission using state $S_{n}$. The optimal policy calculator obtains the optimal action of (26) from the channel estimates and APP from the data detector. The state elements are then refined based on the obtained optimal action, and the refined state is used to estimate the channel and optimal policy for the next step.

**Application of other data detection:** The proposed RL-aided channel estimator can be universally applied to any other soft-output data detection method. To achieve this, the proposed RL-aided channel estimator relies on the availability of APPs, which can be directly derived from the MAP data detection method. In the case of using other soft-output data detections, the proposed RL-aided channel estimator can utilize the APPs that are computed from the log-likelihood ratios.

**Complexity analysis:** Complexity is analyzed in terms of real multiplications to provide an implementation perspective. Figure 3 shows the hardware structure of the proposed RL-aided channel estimator, which consists of the LMMSE channel estimator, state element refinement, and optimal policy calculator. Because the exact complexity can vary depending on the implementation details, the complexity order (O(·)) of each component is analyzed.

The complexity order of the LMMSE channel estimator in (7) is $O((N_{p}+|C\left(a^{★}\right)\left|\right)$$\left(N_{t}^{2}+N_{t}N_{r})\right)$ where $C(a^{★})$ is the set of selected data symbol vectors. The complexity order of state element refinement in Section 4.1 is $O(4(N_{p}+|C\left(a^{★}\right)\left|\right)$. The complexity of the optimal policy in (25) is primarily determined by the computation of $Q^{\left(a\right)}$. Consequently, the complexity order of the optimal policy in (25) is $O\left(2N_{t}^{2}T_{d}^{2}\right)$. It is important to note that among the components of the proposed channel estimator, the optimal policy calculator has the highest complexity because it performs every data symbol index *n*, while the other components perform every data block index *d*.

## 5. Simulation Results

This section presents the effectiveness of the proposed channel estimator using simulations. The numbers of transmit and receive antennas used were $N_{t}=2$ and $N_{r}=4$. The transmission frame consisted of one pilot block with $N_{p}=8$ symbols and $M_{d}=20$ data blocks with $N_{d}=128$ symbols. Each symbol used 4-quadrature amplitude modulation (QAM) symbol mapping. We adopted turbo channel code with a rate of 1/2 and 16 cyclic redundancy check bits. For the proposed channel estimator, the window size was set to $N_{w}=2\times N_{d}$. The signal-to-noise ratio (SNR) was defined as $E_{b}/N_{0}=1/\left(\log_{2}\left|X\right|\sigma_{n}^{2}\right)$ under the power constraint $E\{∥x\left[n\right]{∥}^{2}\}=1$. The proposed channel estimator was compared with the following methods.

PCSI: This method is ideal for time-invariant channels in which a perfect initial channel estimate is available at the receiver. Because the initial channel changes during data transmission, it is not optimal for time-varying channels.Pilot: This method uses a conventional pilot-aided channel estimator using (3).Soft: This method is a data-aided channel estimator when all symbols in (20) are used as additional pilot symbols.Conv-RL [31]: This method is a data-aided channel estimator in which the detected data symbol is selected using the RL approach developed for time-invariant channels.

The performance of the methods was compared with that of the proposed channel estimator in terms of the block-error rate (BLER) and normalized MSE (NMSE). In addition, we considered the time-invariant channel ϵ=0 and time-variant channel with ϵ=0.005 and ϵ=0.01. Note that channel was more severely variant when ϵ=0.01 than when ϵ=0.005.

Figure 4 shows the BLERs for the proposed and other channel estimators in the time-invariant channel, i.e., ϵ=0. The conventional pilot-aided channel estimator exhibited a poor performance when the number of pilots was small. Data-aided channel estimators can overcome performance degradation caused by pilot-aided channel estimators. In particular, the RL-based channel estimator [31] showed an outstanding performance compared with other channel estimators. The BLER of the proposed channel estimator was slightly worse than that of [31] because of a reduced window size $N_{w}=2\times N_{d}$.

In Figure 5, the proposed channel estimator is compared with other channel estimators in time-varying channels. The proposed channel estimator had a better BLER improvement than the conventional pilot-aided channel estimator. In particular, the performance improvement is more prominent at ϵ=0.01 than that at ϵ=0.005. This is because the proposed channel estimator can efficiently capture channel variations by selecting and refining detected data symbols. In addition, in time-variant channels, the proposed channel estimator had a slightly higher BLER than the RL-based channel estimator, primarily due to the utilization of a reduced window size (see Figure 5). However, in time-varying channels, this reduction in window size actually contributed to an improvement in BLER by effectively leveraging the most recent data symbols. Consequently, the proposed channel estimator had a lower BLER compared to the RL-based channel estimator. In Figure 6, we show the BLER of the proposed channel estimator for different window sizes $N_{w}$ in time-varying channels with ϵ=0.01 The BLER of the proposed channel estimator gradually degraded as $N_{w}$ increased. This is because the selected data symbol is undesirable as an additional pilot symbol in fast fading channels; therefore, only the usage of the latest selected data symbol can improve the performance.

To further investigate the effect of window size, we investigated the NMSE of the proposed channel estimator for different window sizes $N_{w}$ at ϵ=0.005 and $E_{b}/N_{0}=-2$ dB (Figure 7). We observed that the NMSE improved until $M_{d}=2$ but decreased as the data block length increased. This is because the old selected data symbol is ineffective for estimating the channel. Thus, when we discard the old data symbol, we can further improve the estimation accuracy (Figure 7).

## 6. Conclusions

A data-aided channel estimator was proposed for time-varying channels, which involves selecting the detected data symbol. To facilitate efficient selection of the detected data symbol, an optimization problem was initially formulated to minimize the channel estimation error. Subsequently, the MDP for this optimization problem was formulated, and its optimal policy was derived using an RL algorithm. In the derivation process, approximations for the transition probability and a first-order Gaussian–Markov process were utilized. To improve estimation accuracy, a state element refinement was introduced to capture the time-varying nature of the channel by incorporating a window size. Simulation results demonstrated that the proposed channel estimator provides similar performance to the conventional RL-based channel estimator in time-invariant channels when ϵ=0, while showing improved performance in time-varying channels when ϵ=0.01 and ϵ=0.005 compared to conventional RL-based channel estimator.

An interesting direction for further research involves optimizing the frame structure in terms of the spectral efficiency. In this study, the frame structure comprises one pilot and *D* data block. The proposed RL-aided channel estimator is applied to the data blocks to capture the time-varying nature of the channel. However, in fast fading channels, it can be challenging for the proposed channel estimator to accurately track channel variations. In such cases, reducing the value of *D* in the frame structure can potentially improve the performance. However, this reduction also leads to a degradation in spectral efficiency. To find an appropriate value for *D* in time-varying channels, an optimization problem that maximizes spectral efficiency while maintaining acceptable performance levels becomes a suitable criterion. To address this, one approach is to first derive the performance of the RL-aided channel estimator. Subsequently, the solution to the optimization problem can be obtained using the derived performance. 

## Figures and Tables

**Figure 1 sensors-23-05689-f001:**
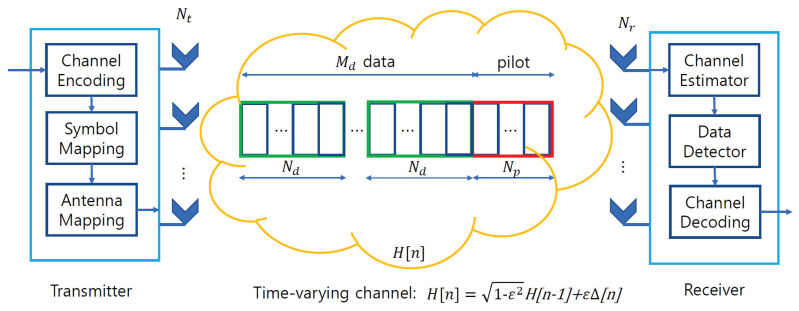
Considered system model and frame structure in time-varying channels.

**Figure 2 sensors-23-05689-f002:**
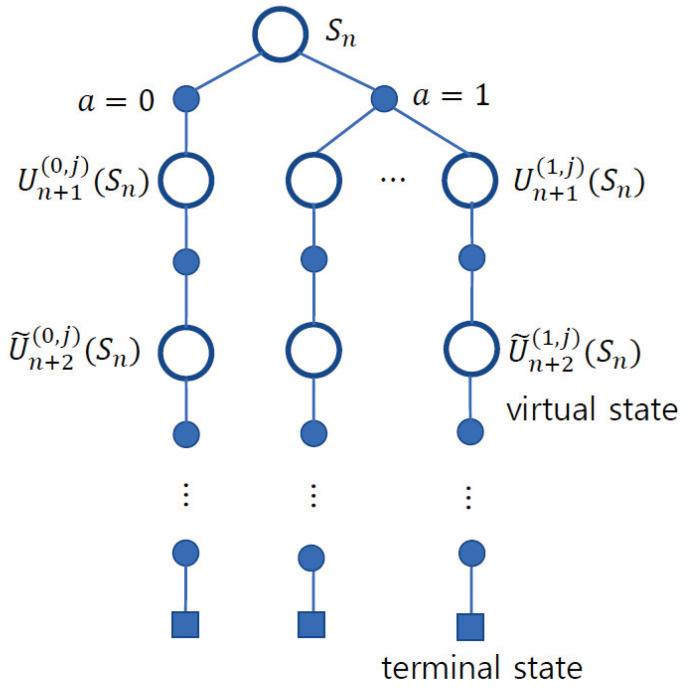
State-action diagram of the proposed channel estimator. After the time slot index n+2, the virtual state is applied such that the state transition is simplified.

**Figure 3 sensors-23-05689-f003:**
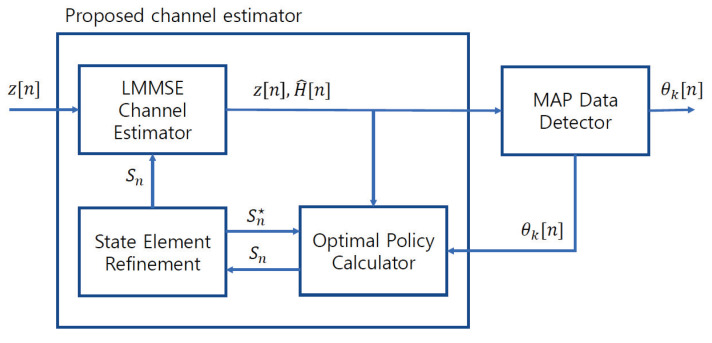
Proposed channel estimator using a further performance improvement strategy.

**Figure 4 sensors-23-05689-f004:**
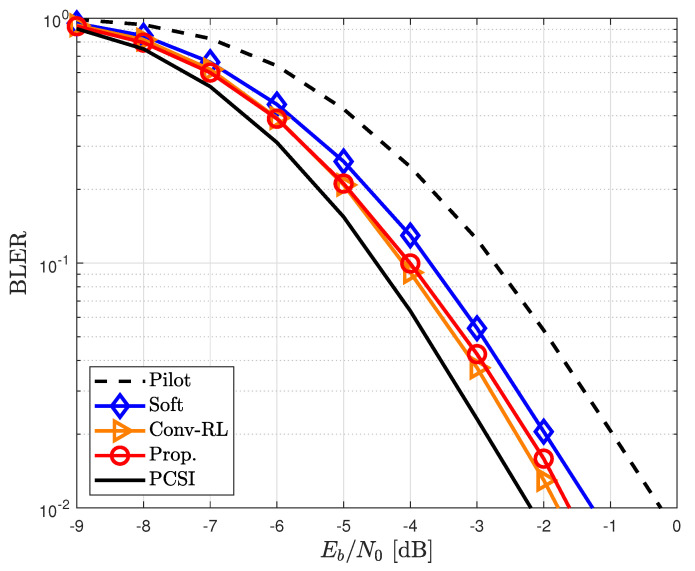
BLER for different channel estimators in time-invariant channels.

**Figure 5 sensors-23-05689-f005:**
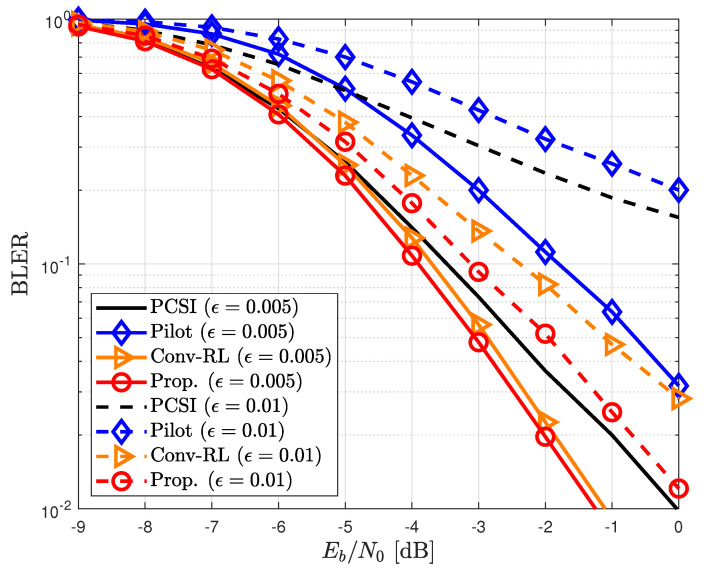
BLER for different channel estimators in time-varying channels (ϵ=0.005 and ϵ=0.01).

**Figure 6 sensors-23-05689-f006:**
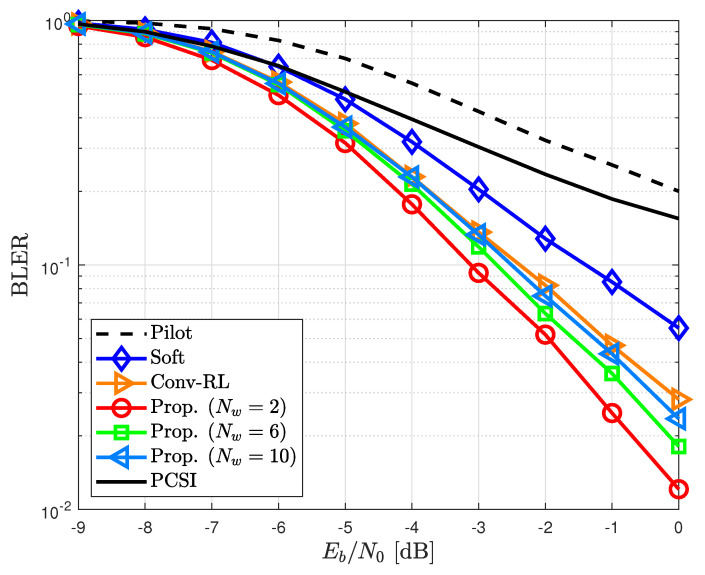
BLER of the proposed channel estimator for different window sizes $N_{w}$ in time-varying channels with ϵ=0.01.

**Figure 7 sensors-23-05689-f007:**
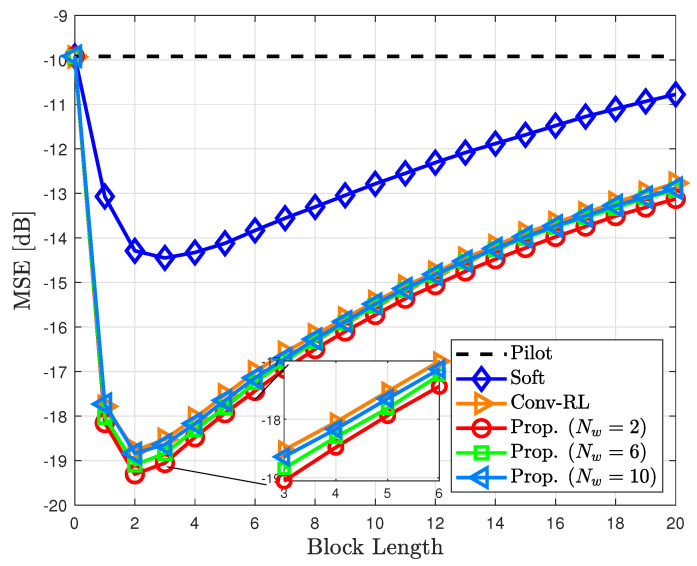
NMSE of the proposed channel estimator for different window sizes $N_{w}$ in time-varying channels with ϵ=0.005.

## Data Availability

Not applicable.

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
