# Peer review of "Reinforcement Learning-Aided Channel Estimator in Time-Varying MIMO Systems"

_sensors, 2023, doi:10.3390/s23125689_

Round 1
Reviewer 1 Report
This paper proposed a reinforcement learning-aided channel estimator for time-varying MIMO systems, where the basic concept is to select the detected data symbol for channel estimation. The reviewer has the following comments:
1. The MAP rule is used for data detection, which achieves good performance but also incurs very high computational complexity. The MAP rule is not often used in practice, how about the performance achieved by other detection algorithms.
2. In practice, pilot signals should be inserted periodically, while in this work, there is only one pilot block, and complex algorithms are used for channel tracking. The performance of such a tracking scheme will degrade significantly when n increases. This part requires further study.
3. Why not use deep reinforcement learning techniques to solve the considered problem?
None
Reviewer 2 Report
the proposed topic and methodology are interesting even if only slightly innovative. The following suggest aimed at improve the quality of the work should be taken into account by the authors.
1) the proposed methodology make use of an optimization algorithm, could you compare the obtained results with other state of the art optimizers?
2) you presented only numerical assessment, I understand that an experimental assessment could be complicate. Could you insert some suggestions relates to how implement an experimental assessments?
3) please slightly improve the reference section.
The quality of english language is acceptable, there are only minor typos and grammar errors.
Round 2
Reviewer 1 Report
No further comments.
None
Reviewer 3 Report
The revised manuscript is fine. The reviewer has no other questions.